# Association of Maternal Plasma Total Cysteine and Growth among Infants in Nepal: A Cohort Study

**DOI:** 10.3390/nu12092849

**Published:** 2020-09-17

**Authors:** Nikhil Arora, Tor A. Strand, Ram K. Chandyo, Amany Elshorbagy, Laxman Shrestha, Per M. Ueland, Manjeswori Ulak, Catherine Schwinger

**Affiliations:** 1Centre for International Health, Department of Global Public Health and Primary Care, University of Bergen, 5020 Bergen, Norway; docnikhilarora@gmail.com; 2Centre for Intervention Science in Maternal and Child Health, Centre for International Health, Department of Global Public Health and Primary Care, University of Bergen, 5020 Bergen, Norway; tor.strand@uib.no (T.A.S.); manjeswori@gmail.com (M.U.); 3Department of Research, Innlandet Hospital Trust, 2609 Lillehammer, Norway; 4Department of Community Medicine, Kathmandu Medical College, Kathmandu 44600, Nepal; rkchandyo1@gmail.com; 5Department of Physiology, Faculty of Medicine, University of Alexandria, Alexandria 21131, Egypt; amany.elshorbagy@alexmed.edu.eg or; 6Department of Pharmacology, University of Oxford, Oxford OX13QT, UK; 7Department of Child Health, Institute of Medicine, Tribhuvan University, Kathmandu 44600, Nepal; laxmanshree12@gmail.com; 8Department of Clinical Science, University of Bergen, 5020 Bergen, Norway; per.ueland@ikb.uib.no

**Keywords:** amino acid, metabolism, tCys, malnutrition, weight, length, child, stunting, wasting, Asia

## Abstract

Cysteine is a semi-essential amino acid that has been positively associated with growth in children. However, transgenerational effects remain unclear. The aim of this analysis was to assess whether maternal plasma total cysteine (tCys) concentration is associated with various growth indicators in infants living in peri-urban settings in Bhaktapur, Nepal. We used data from the 561 mothers enrolled in an ongoing randomized controlled trial. We built linear regression models to evaluate the associations between maternal tCys and birth weight, length-for-age Z-scores (LAZ) and weight-for-length Z-scores (WLZ) at birth and six months of age. Maternal tCys was inversely associated with birth weight among boys after adjusting for confounders (*p* < 0.05). In addition, there was a negative association between maternal tCys and LAZ at birth (*p* < 0.01). No associations between maternal tCys and the other anthropometric indicators were found significant, although there was a tendency for maternal tCys to be associated positively with WLZ at birth among girls (*p* < 0.10). This is a first study evaluating transgenerational relation of tCys on growth in infants. Further, larger and more comprehensive studies are needed to determine if and how maternal tCys alters child growth.

## 1. Introduction

Malnutrition is a wide-spread health problem leading to profound short- and long-term consequences for child growth and development [1,2,3], as well as for survival [4]. It is estimated that more than 45% of all deaths globally among children under the age of five years have malnutrition as an underlying risk factor, and it is a leading cause of the burden of disease in the world [5]. 

Evidence from several studies on fetal and child growth has shown the importance of the first 1000 days—starting from conception up to the second year after birth [2,6]. A significant proportion (20–30%) of wasting and stunting is found to originate in utero [7]. These observations reinstated the fact that the nutritional status of a pregnant woman is not only relevant for her own health but also affects her growing fetus/child. 

Although there are many risk factors associated with malnutrition [2,8,9,10], a diverse diet, rich in all necessary nutrients, is very important, especially among vulnerable populations such as growing children and pregnant women [11]. High quality protein intake has been shown to promote child growth [12], possibly via enhancing insulin-like growth factor-1 (IGF-1) production [13]. 

Amino acids are the building blocks of proteins. Deficiency of amino acids was found to suppress cell and organismal growth via the mechanistic target of rapamycin complex 1 (mTORC1) sensing pathway [14]. Cysteine, a sulfur containing proteinogenic amino acid, controls structure and stability of proteins [15] and is also the limiting precursor of the major intracellular antioxidant glutathione (GSH) [16]. Cysteine is considered essential for newborns because of an immaturity of the enzyme *cystathionase* (or *cystathionine γ-lyase*) which is required for the final step of transsulfuration pathway in the conversion of methionine and serine to cysteine [17,18].

N-acetyl cysteine (NAC) is a supplement and precursor of cysteine shown to improve placental functions in various animal studies by upregulating placental antioxidant activity and placental growth factors, thus reducing placental oxidative stress [19,20,21]. NAC supplementation ameliorated intrauterine growth restriction in a study on guinea pigs [19] and cadmium-induced fetal growth restriction in a study on mice [20]. The anti-oxidative properties of NAC have led to its use as prophylaxis to prevent premature birth and recurrent pregnancy loss in pregnant women [22].

Plasma total cysteine (tCys) was associated positively with anthropometric status in a study among 6–30 months old Indian children [23], however the possible influence of maternal tCys on growth among infants during early childhood was not studied. Despite a strong correlation shown by Küster et al. [24] between maternal tCys concentration and cysteine (and GSH) concentration in the offspring, there has been very little emphasis on transgenerational influence of tCys on growth in infants. With this study, we assessed how maternal tCys concentration is associated with postnatal anthropometric status in infants in Bhaktapur, Nepal.

## 2. Materials and Methods

### 2.1. Original Study

This was a secondary analysis of data from an ongoing randomized controlled trial (RCT) registered at www.clinicaltrials.gov (ID NCT03071666), which is taking place in Bhaktapur, Nepal. The trial aims to measure the effect of vitamin B_12_ supplementation during pregnancy and postpartum on growth and neurodevelopment in early childhood. Details on the original study protocol have already been published [25]. In brief, the trial will enroll 800 pregnant Nepalese women (not later than 15 weeks of pregnancy) aged 20–40 years old and residing in Bhaktapur municipality and surrounding areas. Exclusion criteria include no informed consent, taking dietary or multi-vitamin supplements containing vitamin B_12_, known cases of any chronic disease under treatment (such as tuberculosis, diabetes, hypertension, hypo- or hyperthyroidism, pernicious anemia and Crohn’s disease) or current users of anticonvulsant drugs, severe anemia (hemoglobin concentration <7 g/dL), suffering from any condition that requires treatment with vitamin B_12_ and strict vegans. In addition to vitamin B_12_ (50 μg) or placebo, all pregnant women are also given folic acid (0.4 mg) for the first 2 months of pregnancy followed by iron (60 mg elemental iron) and calcium supplements (500 mg) until 45 days after delivery according to WHO guidelines.

Ethical approval for the trial was obtained from Nepal Health Research Council (NHRC; registered number 253/2016) and the Regional Committee for Medical and Health Research Ethics of Western Norway (REK vest; reference number 2016/1620). This study is conducted in accordance with the Declaration of Helsinki.

### 2.2. Laboratory Assessment and Anthropometric Measurements

Maternal blood samples (3 mL) were collected into vials containing ethylenediaminetetraacetic acid (EDTA) as anticoagulant, at the time of enrollment into the trial. The plasma was centrifuged at approximately 700× *g* at room temperature for 10 min, separated and transferred into storage vials and stored at −70 °C until analyses. Plasma total cysteine (tCys) concentration was measured using a modified gas chromatography-mass spectrometry method based on methylchloroformate derivatization having a reported coefficient of variance (CV) of 1–2% within-day and 4–8% between-day [26]. Concentrations of plasma cobalamin (or vitamin B_12_) and plasma folate were estimated by microbiological assays using a chloramphenicol-resistant strain of *Lactobacillus casei* and colistin sulfate-resistant strain of *Lactobacillus leichmannii*, respectively, having a reported CV of 4% within-day and 5% between-day, both for cobalamin and folate [27,28]. Maternal anthropometric measurements were taken by our study staff at the time of screening for the trial and was verified by a supervisor at the time of enrollment of mothers into the trial. The weighing scale (Salter/HoMedics, Kent, UK and Seca, Hamburg, Germany) and stadiometer (Prestige, Hardik MediTech, Delhi, India) used had a precision of 100 g and 1 mm, respectively, and were calibrated regularly. We calculated body mass index (BMI) as the ratio of body weight (in kg) and height squared (in m), expressed as kg/m^2^. 

Birth weight of infants was measured and recorded by hospital staff. Copies of the original records were gathered by our study staff and the birth weight was noted in our case report forms. Anthropometric measurements of the infants were taken in their homes immediately after birth and at six months of age by the study staff. Each infant’s length and weight were measured at least twice, and the average of the two measurements were calculated and used for our analyses. Length was measured according to standard guidelines using portable board (Seca, Hamburg, Germany). Weight was measured with portable electronic scale (Seca, Hamburg, Germany) that measures to the nearest 0.01 kg. 

The study staff had received training before initiation of the trial and supervisors monitored all fieldwork activities, verifying 5% of measurements taken by field workers. Data were double entered into a database and checked for consistency by a supervisor. 

### 2.3. Data Management and Analysis

Statistical analyses were done using Stata 16.0 (Stata Corp. 2019, College Station, TX, USA) and R version 3.6.2 (R Foundation for Statistical Computing, Vienna, Austria). All analyses in the present study were restricted to the group of 561 mother–infant dyads where maternal tCys concentration and anthropometric measurements of their children at birth and six months of age were available. 

Birth weights gathered from hospital records were in grams and used as such in our analyses. Z-scores for length-for-age (LAZ) and weight-for-length (WLZ) for infants were calculated according to WHO Child Growth Standards [29]. A WAMI-index was calculated to represent household socioeconomic status (SES) using the indicators: water and sanitation access, household wealth (assets), maternal education and income. Calculations were adapted from Psaki et al. [30] with each of the indicators equally contributing to the index. The WAMI-index is between 0 and 1 with a higher index indicating a higher SES. Mean (SD; standard deviation) or median (IQR; inter-quartile range) were calculated for continuous variables and proportions for categorical variables. 

We built linear regression models to assess the association between maternal tCys concentration (predictor variable) and birth weight or anthropometric status (LAZ and WLZ scores) in infants at birth and at six months of age (predicted variables). Birth weight, LAZ, WLZ and maternal tCys concentration were used as continuous variables. As there is no established cut-off for tCys, we additionally categorized maternal tCys into <25th percentile, 25th–75th percentile and >75th percentile and used this variable in separate models. The categorical analyses were done to be able to better communicate our results across both ends of maternal tCys, as studies have shown negative growth outcomes with both low and high cysteine concentrations [23,31,32]. We performed purposeful selection of covariates for adjustment into our models [33]. The covariates that we checked for as our potential confounders in each model include maternal age, body mass index (BMI), parity, years of education, household SES, infant’s gender, maternal plasma cobalamin and folate concentrations. First, univariate analyses of each potential covariate were done, using the significance level of 0.25 as a screening criterion for initial variable selection. Second, a multivariable analysis with the selected covariates for each predicted variable was done. All covariates not significant at a level of 0.05 were taken out one by one from the multivariable model and removed if the coefficient for predictor variable did not change by ≥15%. Third, each of the covariates that had been screened out in Step 1 were added back to the models one by one and retained if it changed the coefficient of the predictor variable by ≥15%. Lastly, each of the final models were checked for its interaction by gender. A *p*-value ≤ 0.05 was considered statistically significant.

In addition, generalized additive model (GAM) analyses were performed to explore any non-linear associations between maternal tCys and birth weight or anthropometric indices (LAZ and WLZ) at birth and at six months of age. All confounders identified in the linear regression models were adjusted for and values <2.5th percentile and ≥97.5th percentile for maternal tCys were excluded to avoid overfitting at the extremes.

## 3. Results

### 3.1. Population Characteristics

Maternal characteristics were available for 561 enrolled mothers, and birth weight and anthropometric measurements at birth were available for 521 infants. The data for the current analyses were obtained from the ongoing intervention study; therefore anthropometric measurements at six months were available for 376 infants (Appendix A). Demographic, nutritional and socioeconomic indices of the available mother–infant dyads are summarized in Table 1. On average, mothers were 27.5 years old at enrollment, with 48.0% (269) mothers being nulliparous. The mean (±SD) maternal tCys concentration was 207.3 (±24.6) μmol/L. The median age (IQR) of infants at two anthropometric assessments by study staff was 3 (2–5) and 182 (181–184) days. For 18 observations, WLZ score at birth was outside the reference range of WHO Child Growth Standards. There was an improvement in mean LAZ and WLZ scores for infants at six months of age relative to assessment done at birth.

### 3.2. Maternal Cysteine and Infant Growth

Table 2 shows the association between maternal tCys concentration and the selected growth indicators. For the birth weight model, the interaction between maternal tCys and gender was significant and therefore reported separately. Maternal tCys concentration was inversely associated with birth weight among boys (β = −2.611, 95% CI: −4.547, −0.676) after adjusting for relevant confounders and the interaction term (*p* < 0.05). The birth weight decreased on average by approximately 105 g among boys across the highest quartile compared to middle half of maternal tCys concentration (Appendix A). Maternal tCys concentration was also negatively associated with LAZ score at birth (β = −0.005, 95% CI: −0.009, −0.001) (*p* < 0.01). However, stratified analyses revealed that the association reached statistical significance among boys only (Appendix A).

Multivariable linear regression models did not show any significant associations between maternal tCys concentration and WLZ score at birth, LAZ score at six months of age or WLZ score at six months of age (Table 2). However, there was a tendency for maternal tCys concentration to be positively related with WLZ at birth among girls (*p* < 0.10) (Appendix A). Results from multivariable linear regression models using maternal tCys as categorical variable can be found in Appendix A.

Figure 1 shows GAM plots for the relation between maternal tCys and selected growth indicators after excluding values <2.5th percentile and ≥97.5th percentile for maternal tCys. There was a slight indication of an inverted U-shaped relation for all anthropometric indicators, except for WLZ at six months of age. The slopes were more distinct for higher maternal tCys concentration in the models with birth weight (in boys) and LAZ at birth as outcomes.

## 4. Discussion

The current analyses were undertaken to elucidate whether maternal tCys can have an impact on growth in infants until six months of age. In the past, there has been much debate regarding use of cysteine as a supplement to promote growth among premature infants and undernourished children, but the effects of cysteine have been found to be inconclusive [34,35,36]. A recent study done on 2102 children aged 6–30 months in New Delhi, India showed that tCys was positively associated with height-for-age Z-score (HAZ) and weight-for-height Z-score (WHZ) [23]. Although a correlation between maternal tCys concentration and cysteine concentration in the offspring has been shown [24], it is not clear whether maternal tCys has any influence on growth in infants. 

Our study indicated that maternal tCys was inversely associated with birth weight among boys and LAZ at birth. For each unit increase in maternal tCys (in μmol/L) the birth weight among boys and LAZ at birth decreased by 2.6 g and 0.005 Z-scores, respectively. This negative association was more pronounced across the highest quartile of maternal tCys concentration, where the birth weight decreased on average by >100 g among boys compared to middle two quartiles. On stratification by gender, the latter association was found significant among boys only. We did not find any statistically significant associations between maternal tCys and birth weight among girls or WLZ at birth or WLZ/LAZ at six months of age.

There is no established reference range for tCys concentrations in adults [37]. Although studies in the past used different techniques to evaluate tCys [37,38,39], they all found cysteine to be the most abundant aminothiol in healthy subjects, with total concentration approximately 250 μmol/L [40]. A cross-sectional study involving 8585 healthy women in the Hordaland county of Western Norway has shown the mean (2.5–97.5 percentile) tCys concentration to be 253.1 (202.1–317.1), 275.8 (215.4–347.2) and 296.3 (233.5–360.5) μmol/L for women aged 40–42, 43–64 and 65–67 years, respectively [41]. In our study, we observed mean (±SD) tCys concentration in mothers enrolled to be 207.3 (±24.6) μmol/L. This relatively lower tCys concentration might be due to higher utilization of cysteine [42], hemodilution effect during pregnancy [43], potentially younger age of our enrolled population relative to other studies [41] or low intake of animal products commonly seen among women of reproductive age in Nepal [44].

Our study used data from an ongoing RCT where women in the intervention arm were given 50 μg of vitamin B_12_ [25]. The maternal blood samples for tCys evaluation in our analysis were taken at enrollment of mothers into the original study and prior to supplementation with vitamin B_12_ or placebo. A recent meta-analysis of 18 studies with 11,216 observations showed no linear association of maternal vitamin B_12_ with birth weight, but maternal vitamin B_12_ deficiency was associated with increased risk for infants born with low birth weight [45]. The effect of vitamin B_12_ supplementation during pregnancy or postpartum on growth outcomes in early childhood has to date not been established; if there would be an effect, this could have diluted our results. Cobalamin (or vitamin B_12_) and folate are important cofactors involved in the re-methylation of homocysteine to methionine, and their deficiencies are associated with elevated plasma homocysteine levels and reduced transsulfuration [15,46]. Thus, both maternal plasma cobalamin and folate concentrations were examined as potential confounders in our models. 

### 4.1. Maternal Cysteine and Birth Weight

In the study on newborns by Küster et al. [24], the mothers to infants born preterm were found to have low cysteine levels, possibly because of a higher requirement of cysteine due to oxidative stress associated with prematurity. In animal studies, supplementing NAC during pregnancy prevented pregnancy related complications probably through its placental anti-oxidative effect [19,20,21]. This is also one of the reasons for its use as a prophylaxis to prevent premature birth and recurrent pregnancy loss in pregnant women [22]. Nonetheless, a large study by El-Khairy et al. [32] assessed the outcomes of 14,492 pregnancies among 5883 women in Norway and showed that high maternal tCys concentration (tCys ≥ 304 μmol/L) was associated with higher risks of preeclampsia, premature delivery and very low birth weight, even though the maternal tCys was measured years after the outcomes had occurred. Our findings predict a significant inverse association between maternal tCys and birth weight among boys where, with each unit of maternal tCys (in μmol/L), the birth weight decreased by 2.6 g despite low overall maternal tCys concentration in our study population. The mechanism remains unclear and the difference in gender has not been observed before. Indeed, elevated tCys is speculated to provoke placental vascular dysfunction due to its effect on endothelial function, which in turn may cause various pregnancy related complications [47]. It is still not clear if the proposed effect of elevated tCys on placental vasculature is an independent effect or an artifact stemming from elevation in plasma total homocysteine (tHcy) concentration which is strongly associated with tCys [47,48,49]. El-Khairy et al. [32] were also not able to rule out the possibility of interaction between tCys and tHcy in the pathogenesis of pregnancy related complications in their study. In addition, most pregnancy related complications are in fact argued to be strongly interrelated, thus preventing disentanglement of the primary pregnancy related complication caused by elevated tCys [32]. It has been argued that a male offspring is potentially more vulnerable to environmental assaults during pregnancy compared to females [50], which could partly explain our findings.

### 4.2. Maternal Cysteine and Linear Growth

Cysteine supplementation in mice resulted in increased growth plate thickness through upregulation of IGF-1 [31], suggesting that cysteine supports linear growth. Despite the evidence from this animal study, the role of cysteine on linear growth in humans has not been investigated. Schwinger et al. [23] demonstrated a positive association between tCys and HAZ in children at 6–30 months of age, but they did not investigate the possible influence of maternal tCys on linear growth in infants at birth and thereafter. The present study is the first study to evaluate the association between maternal tCys concentration and linear growth during infancy. In conflict with findings shown in mice [31] and by Schwinger et al. [23], we found a negative association between maternal tCys and linear growth during early infancy (i.e., at birth), although this seemed more pronounced in the upper range of maternal tCys, especially in boys. The mechanism underlying this association is not known. Our findings are reminiscent of a multi-ethnic Asian population study conducted in Singapore that has shown that lower protein and higher carbohydrate or fat intake during 26–28 weeks of gestation was associated with longer birth length among boys [51]. A postulated reason for the difference based on gender was that the male offspring tend to grow faster inside the womb, making them more vulnerable in circumstances of maternal nutritional imbalance during pregnancy compared to female offspring [50,52]. 

### 4.3. Maternal Cysteine and Ponderal Growth

Studies on cysteine in children with severe acute malnutrition suggested reduced cysteine production [53] and greater dietary cysteine requirements [54] to combat for oxidative stress, poor immune response and impaired gut function. Indeed, a study in preterm infants showed positive effects of supplementing cysteine enterally on weight gain [34]. A recent study by Schwinger et al. [23] found a positive association between tCys and WHZ in children at 6–30 months of age. Moreover, human [55,56] and rodent [57] studies exhibited that tCys concentration was positively and independently associated with body fat mass and obesity. Elevated cysteine was suggested to be a cause rather than a consequence of obesity, by promoting lipogenesis, inhibiting lipolysis, decreasing energy expenditure and decreasing insulin sensitivity, thus favoring lipid storage via unspecified pathways [58]. However, tCys in children aged 4–19 years was significantly associated with body fat only within an overweight/obese subgroup but not in normal weight children [59]. None of these studies have taken into account the possible transgenerational influence of tCys on ponderal growth in infants at birth and thereafter. Our findings fail to show any significant associations between maternal tCys and WLZ at birth or six months, suggesting a lack of any positive transplacental effect of maternal circulating tCys on offspring body weight.

### 4.4. Limitations and Strength

Our analyses were based on the sample size of 521 and 376 infants at birth and six months of age, respectively. The moderate sample size might have limited our statistical power to detect an association. This research is a prospective cohort study. Data come from an RCT which might limit the generalizability of our findings. Although we adjusted our models for several relevant confounders, we still cannot exclude residual confounding. According to the recent reports by Nepal Demographic and Health Surveys (NDHS) 2016, 13.5% and 15.2% of infants <6 months were stunted and wasted, respectively, compared to 17.6% and 21.3%, respectively, of infants aged 6–8 months [60]. In our study population, 12.3% and 12.5% of infants were stunted and wasted, respectively, at birth compared to only 5.3% and 1.9%, respectively, at six months of age. This relatively lower prevalence of stunting and wasting in our population might be because of peri-urban setting of our study. However, we do not expect the lower stunting and wasting prevalence would have affected our findings.

All pregnant women enrolled were given iron, folic acid and calcium supplements as per WHO guidelines. We cannot exclude if these co-interventions would have affected our findings. In addition, we limited our research by focusing on measures of maternal total plasma cysteine concentration, and we did not take into account maternal dietary intakes into these analyses. Plasma tCys concentration has been shown to respond to sulfur amino acid (SAA) intake within hours, potentially masking long-term inadequate intakes of SAAs [61,62,63,64]. However, Jones et al. [61] argued that the recent history of SAA intakes will not be sufficient to mask long-term inadequacies. Compared to other studies mostly done in high income countries [32,37,38,39], the tCys concentration in our study population appeared to be on the low side. Cysteine is most abundantly found in animal derived foods, which are often not affordable or not eaten due to cultural reasons among populations in low income countries, such as Nepal [44]. This is supported by a study showing that animal derived protein, especially dairy products, were related to serum tCys, while plant derived proteins were not [64]. The use of high-quality data collected under supervision by trained field workers was one of the strengths of our study.

## 5. Conclusions

Nepalese women with a high plasma total cysteine concentration had an increased tendency of giving birth to low birth weight and short statured boys, but there was no effect on girls. The mechanism behind these associations and difference based on gender remain undecipherable. However, our findings fit well with the hypothesized effect on low birth weight via vascular dysfunction as well as the observation that short birth length was associated with high maternal protein intake. Larger and more comprehensive studies with those involving detailed dietary nutrient intake evaluation are required to verify these findings in Nepalese and other populations.

## Figures and Tables

**Figure 1 nutrients-12-02849-f001:**
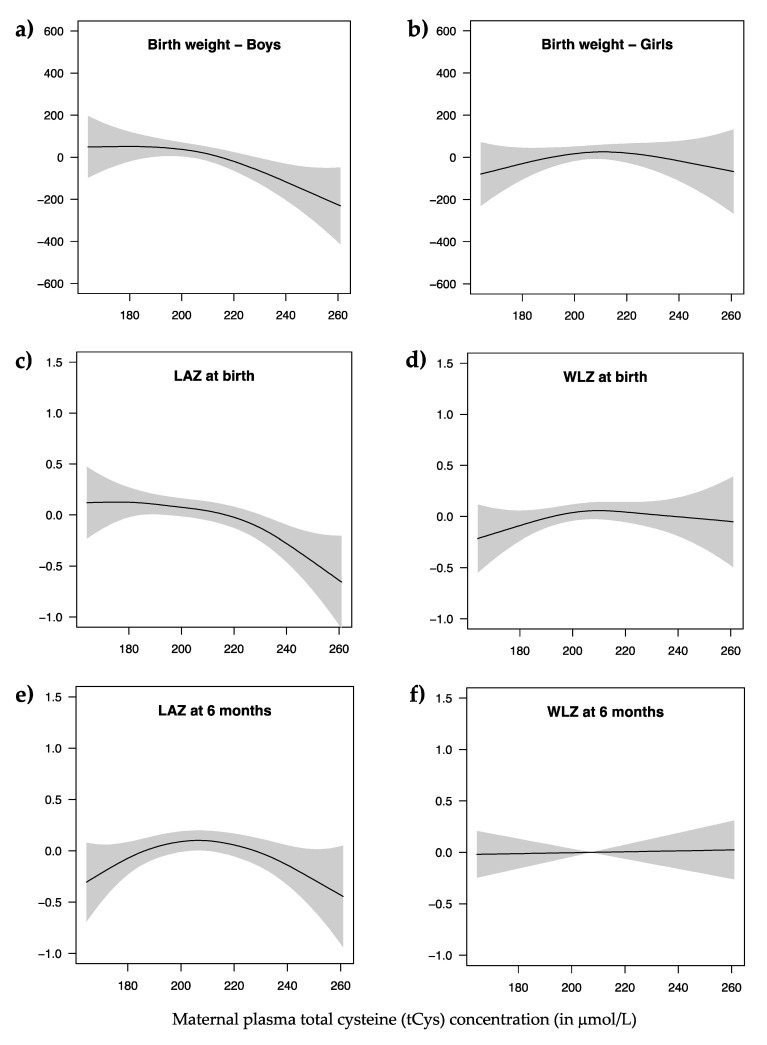
Generalized additive model (GAM) plots showing: the relation of maternal plasma total cysteine (tCys) concentration with birth weight (in grams) among boys (**a**) and girls (**b**); length-for-age Z-score (LAZ) at birth (**c**); weight-for-length Z-score (WLZ) at birth (**d**); LAZ score at six months of age (**e**); and WLZ score at six months of age (**f**) after adjusting for confounders included in the regression models and excluding values <2.5th percentile and ≥97.5th percentile for maternal tCys. Y-axes values for all the growth indicators are centered around their respective median.

**Table 1 nutrients-12-02849-t001:** Characteristics of study population.

Maternal Characteristics (*n* = 561)	Estimate
Mean age (SD), years	27.5 (3.8)
Mean gestational age (SD) at enrollment by LMP ^1^, weeks	10.2 (3.0)
Mean weight (SD), kg	55.3 (7.7)
Mean height (SD), cms	152.8 (5.3)
Mean BMI (SD), kg/m^2^	23.7 (3.0)
Nutritional status, % (*n*) Underweight (BMI < 18.5 kg/m^2^) Normal weight (BMI ≥ 18.5 and < 25 kg/m^2^) Overweight (BMI ≥ 25 and < 30 kg/m^2^) Obese (BMI ≥ 30 kg/m^2^)	1.3 (7) 65.0 (365) 33.0 (185) 0.7 (4)
Parity, % (*n*) 0 ≥1	48.0 (269) 52.0 (292)
Mean education (SD), years	11.0 (3.5)
Mean tCys concentration (SD), μmol/L	207.3 (24.6)
Median plasma folate concentration (IQR), nmol/L	57.3 (33.0–76.4)
Mean plasma cobalamin concentration (SD), pmol/L	204.5 (78.5)
Mean WAMI-index score (SD) ^2^	0.65 (0.14)
**Infant characteristics (*n* = 521) ^3^**	
Gender, % (*n*) Male	53.6 (279)
Mean birth weight (SD) measured at hospital, g	3009 (428)
Median age (IQR) for assessment at birth, days	3 (2–5)
Median age (IQR) for assessment at six months ^4^, days	182 (181–184)
Mean LAZ score (SD) at birth	−0.85 (1.07)
Mean LAZ score (SD) at six months ^4^	−0.56 (0.90)
Mean WLZ score (SD) at birth ^5^	−0.80 (1.11)
Mean WLZ score (SD) at six months ^4^	0.26 (1.03)

^1^ Missing value of one observation, thus *n* = 560. ^2^ Missing values of seven observations for variables used in WAMI-index score calculations, thus *n* = 554. ^3^ Infants reported as dropped-out = 40. ^4^ Out of 521 infants, variables at six months of age were available for 376 infants. ^5^ WLZ score at birth for 18 observations were outside the reference range of WHO Child Growth Standards (*n* = 503). BMI, body mass index; IQR, inter-quartile range; LAZ score, length-for-age Z-score; LMP, last menstrual period; SD, standard deviation; tCys, plasma total cysteine; WLZ score, weight-for-length Z-score.

**Table 2 nutrients-12-02849-t002:** Multivariate linear regression models for anthropometric measurements and maternal plasma total cysteine (tCys) concentration (in μmol/L).

Anthropometric Indices	*n*	Crude β-coefficients (95% CI) for tCys	Adjusted β-coefficients (95% CI) for tCys
Birth weight ^1^, g	521	−1.072 (−2.579, 0.434)	-
Boys		-	−2.611 (−4.547, −0.676) *
Girls		-	0.502 (−1.792, 2.796)
LAZ score at birth	521	−0.005 (−0.009, −0.001) **	−0.005 (−0.009, −0.001) **
WLZ score at birth ^2^	503	0.003 (−0.001, 0.007)	0.003 (−0.002, 0.007)
LAZ score at six months ^3^	376	−0.002 (−0.006, 0.001)	−0.001 (−0.005, 0.003)
WLZ score at six months ^4^	376	0.002 (−0.002, 0.006)	0.001 (−0.003, 0.005)

^1^ Adjusted for maternal BMI, infant’s gender and interaction between tCys and infant’s gender. ^2^ Adjusted for maternal BMI, education and parity. ^3^ Adjusted for maternal BMI, parity, WAMI, plasma cobalamin and folate concentrations (*n* = 371 for adjusted tCys because of missing WAMI-index values). ^4^ Adjusted for maternal BMI, parity and plasma cobalamin concentration. * *p*-value < 0.05. ** *p*-value < 0.01. BMI, body mass index; CI, confidence interval; LAZ score, length-for-age Z-score; WLZ score, weight-for-length Z-score.

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
