# Peer review of "Association of Maternal Plasma Total Cysteine and Growth among Infants in Nepal: A Cohort Study"

_nutrients, 2020, doi:10.3390/nu12092849_

Round 1

Reviewer 1 Report

The main topic of the manuscript is the effect of maternal cysteine concentration on the birth condition of the newborn. A very interesting issue and relatively little described in the literature.

Some comments:

The methodology lacks information on how maternal anthropometry data was collected and calculated. The characteristics of mothers should include information on how many women were malnourished and how many were overweight or obese (according to BMI). Were the measurements of the body weight and body length of the newborns one-time or the average of three measurements? In the methodology, chromatography parameters can be given, the reader should be able to recreate the experiment.

The study lacks information on women's diet and the effect of nutrition on cysteine levels. This is crucial for assessing cysteine levels.

The conclusion is quite modest.

Author Response

Hello,

Thank you for a very nice feedback. Below we have tried to address to all your comments point-by-point. Please follow our responses added with a different colour to your comments.

  1. The methodology lacks information on how maternal anthropometry data was collected and calculated. Details added from lines 104 to 109. 
  2. The characteristics of mothers should include information on how many women were malnourished and how many were overweight or obese (according to BMI). Information added to Table 1.
  3. Were the measurements of the body weight and body length of the newborns one-time or the average of three measurements? Details added to lines 113 and 114.
  4. In the methodology, chromatography parameters can be given, the reader should be able to recreate the experiment. Changes were made from lines 98 to 104 and the required information was added.
  5. The study lacks information on women's diet and the effect of nutrition on cysteine levels. This is crucial for assessing cysteine levels. Information added from lines 307 to 310 and 312 to 316 to address this concern.
  6. The conclusion is quite modest. The analysis is based on observational data and thus, we cannot make inferences on causality. Therefore, we would like to keep the conclusion as it is, i.e. that we saw an association. However, based on previous literature we added potential mechanisms that could explain our findings (see lines 322 to 324).

Your feedback has helped us improve our manuscript. Thank you! We hope that you agree to the responses to all your comments.

With warm regards,

Nikhil Arora (on behalf of all co-authors) 

Reviewer 2 Report

This study is designed well and the report is clearly written, but unfortunately the data analysis does not lead to conclusions that make physiological and developmental sense.

Not clear why there were categorical analysis (supplemental materials).

What does the term “conditional” refer to?

Table 2 indicates a significant effect for LAZ but the text states there was no effect?

The Heading for the columns in Table 2 is not provided, not sure if it is the Beta values. 

The Table 3 is confusing…as the labels are missing on the ordinates

There is not physiological rationale for the findings which are in the reverse direction form what would be expected.  Thus, it cannot be explained and interpreted.  So also is ther gender effect, which is also not explainable

Author Response

Hello,

Thank you for a very nice feedback. Below we have tried to address to all your comments point-by-point. Please follow our responses added with a different colour to your comments.

  1. This study is designed well and the report is clearly written, but unfortunately the data analysis does not lead to conclusions that make physiological and developmental sense. Thank you for this comment which allowed us to improve the formulation of our conclusion. Although physiological mechanisms are not completely understood, we try to give some potential explanations throughout our discussion. We added this to our conclusion to make this even clearer.  

  2. Not clear why there were categorical analysis (supplemental materials). This is now explained from lines 138 to 140.

  3. What does the term “conditional” refer to? The word “conditionally essential” is replaced with word “semi-essential”. It refers to the inability of special groups such as newborns to synthesize cysteine de novo to satisfy their needs. 

  4. Table 2 indicates a significant effect for LAZ but the text states there was no effect? Thank you for pointing this out; however, we are not aware about this contradiction in our text. We would like to emphasize that LAZ was only significantly associated with tCys at birth and not at 6 months of age. Maybe this can have contributed to some confusion. In case we overlook anything in our manuscript we would like to kindly ask you to refer to a line number so that we can change the wording if necessary.

  5. The Heading for the columns in Table 2 is not provided, not sure if it is the Beta values. This has been clarified now after slight changes to the headings for the columns in Table 2 and in supplementary material.

  6. The Table 3 is confusing…as the labels are missing on the ordinates. Here, we believe you mean Figure 1 and not Table 3. We believe that there has been some glitch with the various versions of MS Word that some information went missing. We are requesting the assisting editor to look into the matter. To resolve the issue from our end and to avoid any information to go missing, we are also submitting a pdf-version of our revised manuscript which will help us to convey complete information to the reviewers and the journal.

  7. There is not physiological rationale for the findings which are in the reverse direction form what would be expected. Thus, it cannot be explained and interpreted. To clarify this point we made the following changes to the text. Lines 241 to 245 with reference to El-Khairy study showed similar results in terms of birth weight. This has been further argued in relation to close association of tCys and tHcy in lines 250 to 253 with appropriate references. A study by Chong et.al. conducted in Singapore showed reminiscent findings where low protein intake among mother during gestation leads to longer birth length among boys (see lines 270 to 272). So also is there gender effect, which is also not explainable. The gender effect could potentially be explained by the greater vulnerability of male offspring compared to female offspring in the pregnant uterus as shown by some studies. However, this is a hypothesis only which is discussed in the literature. The information was added to lines 257 and 258 and from lines 272 to 275.

Your feedback has helped us improve our manuscript. Thank you! We hope that you agree to the responses to all your comments.

With warm regards,

Nikhil Arora (on behalf of all co-authors) 

Round 2

Reviewer 2 Report

The reviewers have been responsive to the critiques.

Author Response

Dear Sir/Ma´am,

I have seen that there are no any further comments/suggestions from your end. Thus we are assuming that you have endorsed the revised version of our manuscript. Thank you for your comments/suggestions during the first round. It has greatly helped us improve our manuscript.

Thanks again!

Best regards,

Nikhil Arora (on behalf of all co-authors)